# Are Fat Mass and Lean Mass Associated with Grip Strength in Adolescents?

**DOI:** 10.3390/nu14163259

**Published:** 2022-08-10

**Authors:** Susana Cararo Confortin, Liliana Yanet Gómez Aristizábal, Maylla Luanna Barbosa Martins Bragança, Luciana Costa Cavalcante, Janete Daniel de Alencar Alves, Rosangela Fernandes Lucena Batista, Vanda Maria Ferreira Simões, Poliana Cristina de Almeida Fonseca Viola, Aline Rodrigues Barbosa, Antônio Augusto Moura da Silva

**Affiliations:** 1Department of Collective Health, Federal University of Maranhão, São Luís 65020-905, Maranhão, Brazil; 2Department of Nutrition, Federal University of Piauí, Teresina 64049-550, Piauí, Brazil; 3School of Sports, Federal University of Santa Catarina, Florianópolis 88040-900, Santa Catarina, Brazil

**Keywords:** muscle strength, adolescent, anthropometry

## Abstract

**Background**: The interaction between lean body mass (LBM) and fat mass index (FMI) with grip strength (GS) has not been explored in the same analysis model in adolescents. This study thus aims to analyze the association between FMI and LBM with GS. **Methods**: This cross-sectional study was conducted with data from the 2016 follow-up of the 1997/98 Birth Cohort of São Luís. Grip strength was assessed by the Jamar Plus + dynamometer. The LBM and FMI indexes were assessed [ratio of the mass (lean or fat-kg) to height (m^2^)]. The confounding variables identified for the relationship between FMI and LBM with GS in the same analysis model, by directed acyclic graph (DAG), were sex, age, race, work, alcohol consumption, smoking, physical activity, and consumption of ultra-processed foods and culinary preparations, used in the adjusted analysis. **Results**: A total of 2339 adolescents (52.5% girls) were analyzed. The boys have a higher GS than the girls. In the adjusted analysis, with each increase of 1 kg/m^2^ in the FMI, GS was reduced by 0.72 kgf for boys and 0.35 kgf for girls. At each increase of 1 kg/m^2^ in the LBM, GS increased by 2.18 kgf for boys and 1.26 kgf for girls. **Conclusions**: FMI was associated with lower GS regardless of the LBM. LBM was associated with higher GS regardless of the FMI.

## 1. Introduction

Adolescence is a period of growth, development, and maturation, with changes in body composition different between men and women [1]. These changes result from the dynamic and complex growth process and include alterations in the storage and distribution of muscle, bone, and adipose tissues [2]. Another factor that could affect the amount of muscle mass and strength levels would be the difference between lower plasma concentrations of metabolic hormones (testosterone and GH), which are higher in girls compared to boys [3].

Regarding muscle function, grip strength (GS) is used to help predict an individual’s lifelong health status and identify their development. Evidence shows that lower strength is associated with various diseases, which can increase morbidity and mortality from cardiovascular risk [4,5,6].

Some of its components, especially the amount and distribution of fat and lean mass, are significant for the health of children and adolescents [7]. Moreover, body composition is influenced by several individual factors (genetics, birth weight, breastfeeding, physical activity, and food) [2], family factors (schooling and socioeconomic level, etc.), and geographic factors (place of residence, urban environment, academic performance, etc.) [8].

Although some studies describe a relationship between grip strength and body composition indicators, most of them associate GS with body mass index (BMI) [9,10,11]. We found no studies on the association of GS with lean body mass (LBM) and fat mass index (FMI). A person’s fat mass index is an indicator of how much fat weight the person has relative to their own height [12]. BMI is considered a screening index of body weight, being commonly used in epidemiological research or clinical and public health assessments as a substitute measure of body fat [13]. However, this index does not identify the components of body composition, that differ between the sexes [14,15]. FMI and LBM, which consider the distribution of muscle mass and fat mass, are more accurate measures to distinguish these components [16].

Several studies have assessed grip strength in older adults [3,17,18], but not in adolescents. Therefore, no studies have associated LBM and FMI with GS in adolescents. This study thus aimed to analyze the association of LBM and FMI with GS in a cohort of Brazilian adolescents, assessing its interaction with sex.

## 2. Methods

This is a population-based cross-sectional epidemiological study conducted with data from the second follow-up of a birth cohort conducted in São Luís in 1997/98. Population detailing, sample selection, and site characterization have been published previously [19,20]. At baseline (1997), the sample had 2493 live births. In the first study follow-up, in 2005/2006, 673 children aged 7 to 9 years were re-assessed. In the second follow-up, in 2016, 687 adolescents aged 18 to 19 years were followed-up again. This stage included a retrospective component (with the application of a fundamental part of the perinatal questionnaire to the mothers of adolescents) and added 1828 adolescents born in São Luís in 1997 who did not participate in the original cohort. Therefore, 2515 adolescents participated in the second follow-up [20].

### 2.1. Data Collection

A structured instrument of data collection was used, and the Research Electronic Data Capture (REDCap) was used to record and manage the data collected via face-to-face interviews. 

### 2.2. Grip Strength (Dependent Variable)

Grip strength (GS-kilogram force-Kgf) was measured using the Jamar Plus + dynamometer (Sammons Preston). The instrument was adjusted for each individual according to hand size. The participant should be seated for the assessment, with his feet resting on the floor, elbow in 90 degree flexion, forearm in neutral position, and palm of the hand facing upwards [21]. The individual should then apply the greatest possible grip strength in each of the three measurements in each arm, giving a 1 min break for each, using the average force of the dominant hand in kilogram force (Kgf).

### 2.3. Independent Variables

The assessment indexes of the distribution of body components were the lean body mass (LBM, in Kg/m^2^) and fat mass index (FMI, in Kg/m^2^). Firstly, muscle mass was assessed by dual-energy X-ray absorptiometry (DXA) using the Lunar Prodigy device from GE Healthcare^®^. Then, LBM was estimated by the ratio of muscle mass, in kilos, to height squared, in meters [22].

Fat mass was assessed by air displacement plethysmography using the BodPod^®^ Gold Standard equipment from COSMED. FMI was then estimated by the ratio of fat mass, in kilos, to height squared, in meters [22].

The indexes were chosen to correct the distribution of muscle mass and fat mass in the body by height [12].

### 2.4. Complementary Variables

The study’s complementary variables were: age (in continuous years); sex (male and female); race (white, black, and mixed-race–excluding people of Asian descent); years of schooling (none to 8 years, 9 to 11 years, and 12 years or over); and socioeconomic classification according to 2016 Brazil Economic Classification (CEB) criteria [A/B(B1+B2), C(C1+C2), D/E, with class A/B being the richest and with the highest schooling levels, and classes D/E being the poorest with the lowest schooling levels] [23]. Currently working (yes/no), currently smoking (yes/no), and alcohol consumption (low risk: <8/high risk: ≥8, using the Alcohol Use Disorder Identification Test (AUDIT)] were assessed [24]. The major depressive episode or depression variable (yes/no) was assessed using the MINI Questionnaire (Mini International Neuropsychiatric Interview-Brazilian version 5.0.0-DSM IV) [25]. Total physical activity [26] was assessed by the short version of the International Physical Activity Questionnaire (IPAQ) [27]. (Insufficiently active: <300 min/week; physically active: ≥300 min/week).

Food consumption was classified according to the level of processing in culinary preparations (fresh or minimally processed foods), processed foods, and ultra-processed foods. This level was assessed according to dietary caloric percentage and categorized into thirds [28]. A food frequency questionnaire (FFQ) was used to assess food intake in Brazilian adolescents [29].

### 2.5. Data Analysis

Descriptive analyses were performed for all variables, analyzing absolute frequencies and percentages by sex using the Chi-square test. Outcome means (GS) were compared between groups using the Mann–Whitney Test. The interaction test performed by the “margins” command found interaction between sex and anthropometric parameters related to GS.

The associations between anthropometric parameters (LBM and FMI) and the outcome (GS) were estimated by crude and adjusted linear regression models with an estimate of the beta regression coefficient and a 95% confidence interval (95%CI). The nonparametric method of local regression (LOWESS–locally weighted scatterplot smoothing) was used to graphically assess possible nonlinearities. In cases of linearity, no polynomial terms were added to the linear regression models.

Predicted probabilities were estimated by margins and converted into graphs by Stata’s marginsplot. The interaction coefficient between sexes was estimated [to verify the results for girls, the difference between the β coefficient of the LBM for boys (β = 2.18; CI95%: 1.98; 2.38), and the interaction term (β: −0.92; CI95%: −1.24; −0.60) was estimated, resulting in a coefficient of 1.26].

The minimum set of factors to minimize possible confounding or selection biases in the analysis were determined by Directed Acyclic Graph (DAG) in the DAGitty^®^ program version 3.0 (Figure 1). The basis for elaborating the interrelationships between the variables of muscle mass and fat mass with HGS was the current literature. 

The variables selected for the backdoor criterion were: Age, sex, race, work, alcohol consumption, smoking, physical activity, and food consumption. 

Then, analysis adjusted for the confounding factors (age, sex, race, work, alcohol consumption, smoking, physical activity, food consumption) of the relationship of LBM and FMI with GS was conducted. Those analyses include the listed confounders and also adjust for the other primary variable (LBM or FMI).

### 2.6. Ethical Considerations

The Research Ethics Committee of the University Hospital—UFMA approved the project referring to the 1997/98 birth cohort of São Luís under Opinion No. 1,302,489. The individuals or their guardians signed the informed consent form. All projects meet the criteria in resolution No. 466/2012 of the National Health Council and its complementary regulations.

## 3. Results

The study’s analytical sample included 2339 adolescents (52.5% girls). The mean values of GS were higher in boys (35.2 ± 7.7) than in girls (21.5 ± 4.8) (<0.001). Table 1 shows that boys and girls presented significant differences in age, schooling years, socioeconomic class, work, tobacco use, alcohol consumption, major recurrent depressive episodes, total physical activity, and intake of culinary preparation, processed food, and ultra-processed food. The mean values of LBM and FMI differed between the sexes. Boys had higher values of LBM and lower values of FMI than girls (Table 1).

Table 2 presents the crude and adjusted (Figure 2 and Figure 3) analyses of the associations between the indicators (FMI and LBM) and GS. The crude analysis showed the association of FMI and LBM with GS and its sex interactions. After adjustments for the confounding variables (age, alcohol consumption, food consumption, physical activity, sex, smoking, work, race, and LBM; age, alcohol consumption, food consumption, physical activity, sex, smoking, work, race, and FMI), the associations remained. Thus, with each 1 kg/m^2^ increase in FMI, GS was reduced by 0.72 kgf for boys and 0.35 kgf for girls; for each 1 kg/m^2^ increase in LBM, GS increased by 2.18 kgf for boys and 1.26 kgf for girls [to verify the results for girls, the difference between the β coefficient of the LBM for boys (β = 2.18; 95%CI: 1.98; 2.38) and the interaction term (β: −0.92; CI95%: −1.24; −0.60) was estimated, resulting in a coefficient of 1.26]. In adolescents, FMI was associated with lower GS regardless of the LBM.

## 4. Discussion

To our knowledge, this is the first study to assess the association of FMI and LBM with GS in the same analysis model in adolescents. The results showed the association of FMI and LBM with GS and its interactions according to sex. In both sexes, and especially in boys, the FMI was associated with lower GS whereas the LBM was associated with higher GS.

This association was more intense and almost twice as high in boys as in girls. The FMI was negatively associated with GS regardless of LBM in both sexes, but especially in boys. Boys had less body fat and more muscle mass than girls, corroborating previous studies [30,31]. As fat mass and lean mass are predictors of grip strength and directly related to it [32,33], we can say that boys had higher GS than girls.

Other biological differences support our findings. Boys have greater muscle mass and GS [34], are more physically active [35], and are diagnosed with fewer recurrent major depressive episodes [36]. Evidence shows that boys also consume healthier and less ultra-processed foods than girls [37]. This lower protein and micronutrient intake favors a higher LBM and lower FMI.

The differences in grip strength between the sexes are well established. Omar et al. [38] observed that, especially after becoming 11 years old, boys had higher GS than girls, reaching the peak of maximum strength at 19 years old. The anthropometric data of body mass and height of boys were most associated with GS. Moreover, a study [39] with male adolescents reaffirmed that GS is sexually dimorphic and can predict social behavior considering measures of aggression and social competition in boys.

However, these results partly differ. Valero et al. [40] showed that individuals (9 to 17 years old) with a higher frequency of overweight/obesity (according to BMI) and changes in body fat, skinfolds, and waist and hip circumferences had lower grip strength. On the other hand, individuals who had adjusted components of physical well-being, blood pressure, and body composition indicators had better muscle performance [40] and, therefore, greater GS. 

In adolescence, boys gain lean mass faster and for a longer time than girls. In girls, relative muscle mass decreases from the onset to the end of puberty (80% to 75% of body weight). Despite the absolute increase in girls’ muscle mass, its percentage decreases with increased adipose tissue [41]. Body mass differences between the sexes are manifested at a later stage, at 16–17 years of age, when boys are taller and heavier than girls. In adults, muscle mass increases from 80% to 85% of body weight to 90% [42].

Grip strength is a known indicator of an individual’s muscle strength and physical condition, used mainly among older adults [3,18] and patients at risk of losing muscle mass [43], such as those with Chronic Kidney Disease and those undergoing dialysis [29,38]. Indirectly, GS reflects muscle mass [33,44]. Our results are the first to assess the relationship between LBM and GS regardless of FMI, especially in a younger population. Our findings are relevant since they show the early influence of LBM on higher GS in young people, which could be a future risk factor for higher morbidity and mortality—especially in men.

This study has limitations. Self-reported race, socioeconomic class, family income, and physical activity could result in information biases despite being obtained via validated instruments applied by trained interviewers. Another limitation is due to the cross-sectional design of the study, which limits the interpretation of causative relationships between explanatory variables and the outcome.

On the other hand, this study also has strengths. The research involved population-based data of adolescents from São Luís (Maranhão) collected with high methodological rigor by teams who received adequate field training. Moreover, anthropometric parameters were verified by air displacement plethysmography and double-energy radiological absorptiometry, considered accurate methods to determine body compartments. A conceptual theoretical model (DAG) was also elaborated to identify confounding factors, adjust the analysis, and, thus, avoid spurious associations and estimation errors. Another strength of this study is the use of Stata’s margins post-estimate command, which estimates predicted probabilities using the conditional prediction method.

Furthermore, the study used LBM and FMI as assessment indexes of the distribution of body components. These indexes are considered more accurate and viable measures to distinguish fat and muscle mass, helping identify changes in body composition between the sexes.

## 5. Conclusions

In both sexes, and especially in boys, the FMI was associated with lower GS, whereas the LBM was associated with higher GS. In adolescents, FMI was associated with lower GS regardless of LBM. Targeted interventions should thus help develop and preserve muscle mass and strength and reduce fat mass to decrease the risk of lower GS in adulthood.

Finally, longitudinal studies and interventions are essential to better investigate the relationship between increased FMI and decreased GS and increased LBM, and the influence of FMI on the reduction of GS in the young population.

## Figures and Tables

**Figure 1 nutrients-14-03259-f001:**
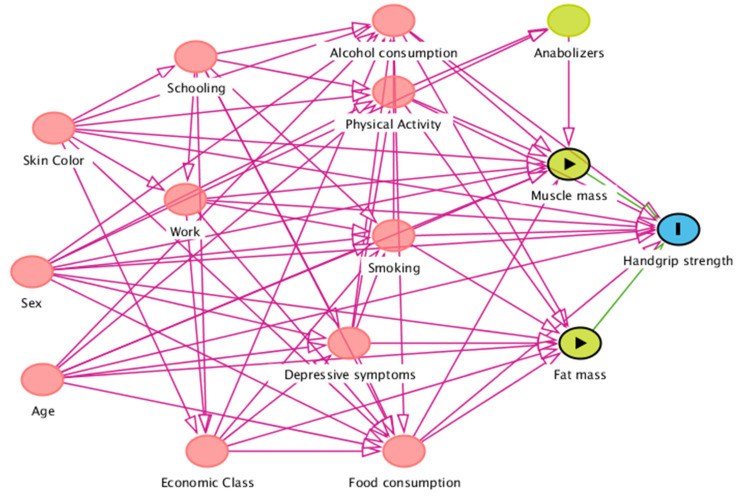
Acyclic graph for the association between muscle mass and fat mass with grip strength.

**Figure 2 nutrients-14-03259-f002:**
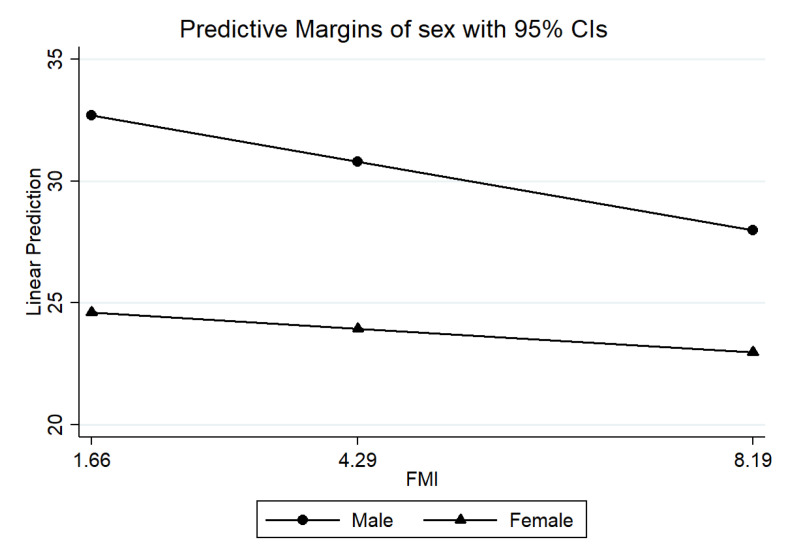
Probabilities predicted by sex compared to fat mass index with grip strength in adolescents. São Luís, Maranhão, Brazil, 2016/2017.

**Figure 3 nutrients-14-03259-f003:**
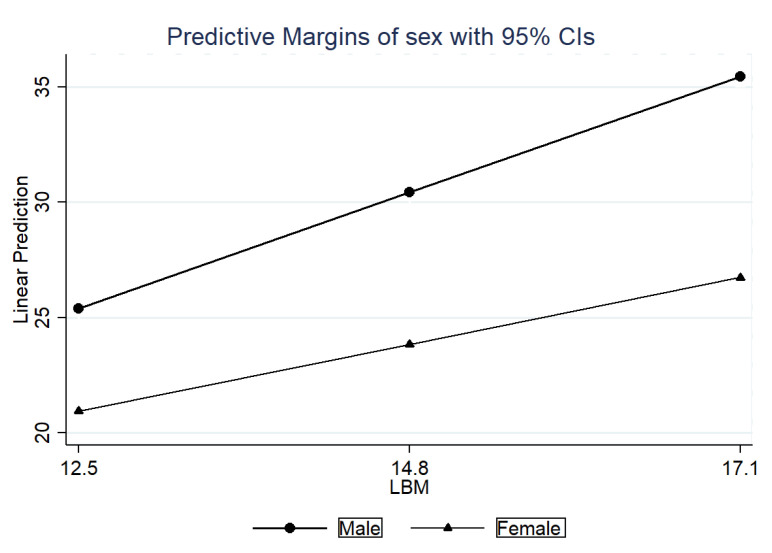
Probabilities predicted by sex compared to lean body mass with grip strength in adolescents. São Luís, Maranhão, Brazil, 2016/2017.

**Table 1 nutrients-14-03259-t001:** Sample characterization according to demographic, socioeconomic, and lifestyle characteristics compared to sex and mean grip strength, by sex, for all variables. São Luís, Maranhão, Brazil, 2016/2017.

	Male		Female			GS		
	*n*	%	*n*	%	*p*	Male Mean (SD)	Female Mean (SD)	*p*
**Age (*n* = 2339)**					0.011			
18	799	49.3	822	50.7		34.9 (7.7)	21.3 (4.9)	<0.001 *
19	313	43.6	405	56.4		36.0 (7.6)	21.8 (4.7)	<0.001 *
**Years of schooling (*n* = 2339)**					<0.001			
0 to 8 years	75	65.2	40	34.8		34.7 (7.8)	21.0 (4.3)	<0.001 *
9 to 11 years	972	47.1	1091	52.9		35.3 (7.7)	21.4 (4.8)	<0.001 *
12 or more years	60	40.0	90	60		34.1 (6.8)	22.1 (5.1)	<0.001 *
**Race (*n* = 2339)**					0.265			
White	205	44.4	257	55.6		34.5 (7.2)	21.3 (4.6)	<0.001 *
Black	180	47	203	53.0		35.1 (8.0)	21.8 (4.8)	<0.001 *
Mixed-race	727	48.7	767	513		35.5 (7.7)	21.5 (4.9)	<0.001 *
**Economic classification (*n* = 2339)**					<0.001			
A-B	312	50.6	305	49.4		35.0 (7.2)	21.5 (4.7)	<0.001 *
C	512	49.2	529	50.8		35.2 (7.7)	21.5 (4.9)	<0.001 *
D-E	151	36.0	268	64.0		36.2 (8.1)	21.4 (5.1)	<0.001 *
**Work (*n* = 2339)**					<0.001			
No	909	46.0	1067	54.0		34.9 (7.6)	21.4 (4.8)	<0.001 *
Yes	203	55.9	160	44.1		36.8 (7.7)	22.1 (5.0)	<0.001 *
**Smoking (*n* = 2339)**					<0.001			
No	1056	46.8	1200	53.2		35.2 (7.8)	21.5 (4.9)	<0.001 *
Yes	56	67.5	27	32.5		35.1 (5.9)	22.0 (3.5)	<0.001 *
**Alcohol consumption (*n* = 2339)**					<0.001			
No	596	43.7	791	56.3		35.6 (7.9)	21.4 (4.9)	<0.001 *
Yes	516	53.0	471	47		34.9 (7.4)	21.5 (4.7)	<0.001 *
**Recurrent major depressive episode (*n* = 2339)**					<0.001			
No	1073	49.6	1088	50.4		35.3 (7.6)	21.5 (4.8)	<0.001 *
Yes	39	21.9	139	78.1		32.4 (8.2)	21.7 (5.4)	<0.001 *
**Total PA (*n* = 2339)**					<0.001			
Insufficiently active	466	35.8	835	64.2		34.1 (7.7)	21.2 (4.8)	<0.001 *
Physically active	646	62.2	392	37.8		36.0 (7.5)	22.0 (4.9)	<0.001 *
**Culinary preparations (%) (*n* = 2339)**					0.006			
1st tertile	337	43.3	441	56.7		34.1 (7.5)	21.2 (4.4)	<0.001 *
2nd tertile	374	47.9	406	52.1		35.1 (7.2)	21.5 (5.2)	<0.001 *
3rd tertile	401	51.3	380	48.7		36.3 (8.2)	21.7 (4.9)	<0.001 *
**Processed food (%) (*n* = 2339)**					<0.001			
1st tertile	322	41.2	459	58.8		35.0 (8.2)	21.5 (4.6)	<0.001 *
2nd tertile	381	48.8	400	51.2		35.1 (7.4)	21.6 (4.9)	<0.001 *
3rd tertile	409	52.6	368	47.4		35.5 (7.5)	21.3 (5.0)	<0.001 *
**Ultra-processed food (%) (*n* = 2339)**					0.001			
1st tertile	406	51.9	376	48.1		36.2 (8.1)	21.5 (5.1)	<0.001 *
2nd tertile	377	48.3	403	51.7		35.5 (7.3)	21.6 (5.0)	<0.001 *
3rd tertile	329	42.3	448	57.7		33.8 (7.4)	21.3 (4.5)	<0.001 *
	** *n* **	**Mean (SD)**	** *n* **	**Mean (SD)**	** *p* **	r (r2) **	r (r2) **	
**FMI (Kg/m^2^)**	1112	3.1 (2.6)	1227	6.5 (3.1)	<0.001	−0.04 (0.16) *p* = 0.122	0.14 (1.96) *p* < 0.001	
**LBM (Kg/m^2^)**	1112	16.4 (1.8)	1227	13.3 (1.6)	<0.001	0.43 (18.49) *p* < 0.001	0.34 (11.56) *p* < 0.001	

Legend: GS—grip strength; LBM—lean body mass; FMI—fat mass index; PA—physical activity; SD—standard deviation. * Wilcoxon test. ** Pearson’s correlation (r = correlation coefficient; r2 = determination coefficient).

**Table 2 nutrients-14-03259-t002:** Crude and adjusted analysis of the relationship of anthropometric parameters associated with GS in adolescents. São Luís, Maranhão, Brazil, 2016/2017.

Variables	Crude Analysis		Adjusted Analysis	
	β (95%CI)	*p*	β (95%CI)	*p*
FMI (Kg/m^2^)	−0.93(−1.03; −0.83)	<0.001	−0.72(−0.86; −0.58) †	<0.001
Interaction (FMI*Girls)	0.27(0.09; 0.44)	0.003	0.47(0.28; 0.67) †	<0.001
LBM (Kg/m^2^)	2.77(2.65–2.89)	<0.001	2.18(1.98; 2.38) ^£^	<0.001
Interaction (LBM*Girls)	−0.81(−1.08; −0.54)	<0.001	−0.92(−1.24; −0.60) ^£^	<0.001

Legend: FMI: fat mass index; LBM: lean body mass; † = Adjusted for age, alcohol consumption, food consumption, physical activity, sex, smoking, work, race, and LBM; ^£^ = Adjusted for age, alcohol consumption, food consumption, physical activity, sex, smoking, work, race, and FMI.

## Data Availability

The data that support the findings of this study are available from e-mail rosangela.flb@ufma.br, but restrictions apply to the availability of these data, which were used under license for the current study, and so are not publicly available. Data are however available from the authors upon reasonable request and with permission of Rosangela Fernandes Lucena Batista.

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
