# Peer review of "Are Fat Mass and Lean Mass Associated with Grip Strength in Adolescents?"

_nutrients, 2022, doi:10.3390/nu14163259_

Round 1

Reviewer 1 Report

Thank you for the opportunity to review this manuscript. The purpose of this study was to analyze the interaction between fat mass index (FMI) and lean body mass (LBM) with grip strength (GS) in a sample of adolescents derived from a birth cohort in Sao Luis from 1997/1998. The authors had a large analytic sample (n=2339) and was able to observe that LBM was positively correlated with GS, while FMI was negatively correlated with grip strength. While this outcome is not necessarily surprising, the study shows strength through the large sample size, inclusion of confounding variables, and sophisticated statistical analysis. However, the study has some significant weaknesses and some oversights that should be addressed. I have provided the following edits and recommendations described below.

Abstract:

The first sentence is somewhat confusing. The authors may want to consider changing it to ‘The interaction between LBM and FMI with GS has not been explored….’

Why do the authors use ‘skin color’ as a descriptive variable instead of ‘race’? This is not typical in research.

The abstract notes that the boys were ‘stronger’ than girls (line 31). Although GS is correlated with strength, this is somewhat of a leap. The authors should just note that the boys have a higher GS.

In the adjusted analysis (line 31-32), it is unclear what the authors are adjusting for. I assume the confounding variables described on lines 28-30 are included, but it is unclear whether FMI was included as a confounder in the LBM to GS analysis, or if LBM was included in the FMI to GS analysis. This is important for the authors to claim that these relationships are independent of each other as suggested in the conclusions sentence.

Introduction:

On lines 43-45, the physiology of why males have more LBM than females is much larger and more complex than what is stated here. I recommend the authors make this more general of a statement.

It is of my opinion that the sentences on lines 50-52 are unnecessary a could be removed.

The authors should describe and expand the evidence and link between GS and overall LBM and strength in adolescents.

The authors should describe what FMI is for the readers. Although BMI is known, FMI is not.

Methods:

The last sentence of the first paragraph (lines 81-82) is repeated in the results section. I recommend the authors only state this one time (methods or results).

Line 92 – Did the participants squeeze the dynamometer continuously for an entire 1 minute?

It is unclear why the researchers did not use the FM from the DXA? DXA provides both overall and segmental LBM and FM. What was the advantage of using the BodPod for FM over the DXA?

One of largest critiques of this study was the data use for adolescents. It is common in research in adolescents to use normative values for many variables including GS, LBM, FMI, and any variable that is affected by growth and development. As the authors know, there is a lot of variability in the growth, development, and maturation of adolescents of the same age and sex. The outcomes of this study are misleading as absolute values of GS, FMI, and LBM are not comparable for adolescents. The data should be re-analyzed using GS percentile/z-score, FMI percentile/z-score, LBM percentile/z-score based on age, sex, height, etc. rather than absolute values. These normative values are available for each of these variables. There are likely Brazilian norms also.

                Grip strength:

                Bohannon, RW., Wang, YC., Bubela, D., Gershon, RC. Handgrip strength: A population-based study of norms and age trajectories for 3- to 17-year-olds. Pediatr Phys Ther. 2017; 29: 118-123.

                Wang YC, Bohannon RW, Li X, Sindhu B, Kapellusch J. Hand-Grip Strength: Normative Reference Values and Equations for Individuals 18 to 85 Years of Age Residing in the United States. J Orthop Sports Phys Ther. 2018 Sep;48(9):685-693.

Body comp (FMI, LBM):

                Weber, D. R., Moore, R. H., Leonard, M. B., & Zemel, B. S. (2013). Fat and lean BMI reference curves in children and adolescents and their utility in identifying excess adiposity compared with   BMI and percentage body fat. The American journal of clinical nutrition98(1), 49-56.

Wells, J. C., Williams, J. E., Chomtho, S., Darch, T., Grijalva-Eternod, C., Kennedy, K., ... & Fewtrell, M. S. (2012). Body-composition reference data for simple and reference techniques and a 4-component model: a new UK reference child. The American journal of clinical nutrition96(6), 1316-1326.

Figure 1 is very impressive

Results:

Line 160 – use ‘GS’ abbreviation

As noted in the abstract comments, it is unclear if FMI was included as a confounder in the LBM to GS analysis, or if LBM was included in the FMI to GS analysis. This issue presents again on lines 177-180. If those analyses include the listed confounders AND also adjust for the other primary variable (LBM or FMI), the authors should make this clear for the reader in the text.

Discussion:

The sentence on line 203-204 is repeated from the previous paragraph. The second sentence of paragraph 2 could start: ‘The association between LBM and GS was more intense…’

The sentence on lines 207-209 is confusing. Please consider re-writing.

Line 206 states that boys had less body fat distributed throughout their body. How can the authors determine segmented body fat distribution if they used a BodPod for determination of FM?

Line 210-211 – As noted previously, the complex physiology leading to increased LBM in males is more complex than just saying ‘because of their testosterone.’ I recommend making these kinds of statements more general.  

The authors state on line 212 that ‘Evidence shows that males consume healthier and less ultra-processed foods than girls.’ The authors collected similar data. Does the author’s data support this?

Line 215-218 should be deleted. The authors did not use normative values for GS, FMI, LBM. Thus, the outcomes of this study, and the interpretations made in these sentences are misleading and incomplete. Strength, LBM is relative to body size, maturational stage, and sex.

Author Response

Response letter

August 2, 2022

Article: ARE FAT MASS AND LEAN MASS ASSOCIATED WITH GRIP STRENGTH IN ADOLESCENTS?

Dear Editor,

We would like to thank you for reviewing and commenting on our article. Suggested changes are answered separately.

The following is a detailed description of how we addressed reviewers’ comments and suggestions.

Reviewer 1

Abstract:

The first sentence is somewhat confusing. The authors may want to consider changing it to ‘The interaction between LBM and FMI with GS has not been explored….’

We appreciate the reviewer’s comment. We changed it as requested.

Why do the authors use ‘skin color’ as a descriptive variable instead of ‘race’? This is not typical in research.

We appreciate the reviewer’s comment. We changed it as requested.

The abstract notes that the boys were ‘stronger’ than girls (line 31). Although GS is correlated with strength, this is somewhat of a leap. The authors should just note that the boys have a higher GS.

We appreciate the reviewer’s comment. We changed it as requested.

In the adjusted analysis (line 31-32), it is unclear what the authors are adjusting for. I assume the confounding variables described on lines 28-30 are included, but it is unclear whether FMI was included as a confounder in the LBM to GS analysis, or if LBM was included in the FMI to GS analysis. This is important for the authors to claim that these relationships are independent of each other as suggested in the conclusions sentence.

We appreciate the reviewer’s comment. We changed it as requested.

Introduction:

On lines 43-45, the physiology of why males have more LBM than females is much larger and more complex than what is stated here. I recommend the authors make this more general of a statement.

We appreciate the reviewer’s comment. Changed the wording of the paragraph so that it would be written in a more general way, as suggested by the reviewers.

It is of my opinion that the sentences on lines 50-52 are unnecessary a could be removed.

We appreciate the reviewer’s comment. We agree with the reviewer, the paragraph was removed.

The authors should describe and expand the evidence and link between GS and overall LBM and strength in adolescents.

We appreciate the reviewer’s comment, but as mentioned in the introduction, although there are studies that describe this relationship between HGS and body composition indicators, most evaluate this association with the Body Mass Index (BMI), in addition, there are no studies that consider the relationship of HGS with specific indicators such as the Lean Mass Index (IMM) and the Fat Mass Index (FMI). This is why it is difficult to add information from studies that analyze this relationship.

The authors should describe what FMI is for the readers. Although BMI is known, FMI is not.

We appreciate the reviewer’s comment. The definition of the FMI was added, as suggested by the reviewers.

Methods:

The last sentence of the first paragraph (lines 81-82) is repeated in the results section. I recommend the authors only state this one time (methods or results).

We appreciate the reviewer’s comment. We changed it as requested.

Line 92 – Did the participants squeeze the dynamometer continuously for an entire 1 minute?

We appreciate the reviewer’s comment. No, the individual makes the greatest possible force, which lasts around 3 seconds, with an interval of 1 minute for each evaluation. We fit in the text.

It is unclear why the researchers did not use the FM from the DXA? DXA provides both overall and segmental LBM and FM. What was the advantage of using the BodPod for FM over the DXA?

We appreciate the reviewer’s comment. We used the Body Fat assessed by BodPod because DXA tends to overestimate body fat when compared to BODPOD1, 2. In addition, we did not study the body segment, but the entire body.

References:

1 - Schubert MM, Seay RF, Spain KK, Clarke HE, Taylor JK. Reliability and validity of various laboratory methods of body composition assessment in young adults. Clin Physiol Funct Imaging. 2019 Mar;39(2):150-159. doi: 10.1111/cpf.12550. Epub 2018 Oct 16. PMID: 30325573.

2 - Lowry DW, Tomiyama AJ. Air displacement plethysmography versus dual-energy x-ray absorptiometry in underweight, normal-weight, and overweight/obese individuals. PLoS One. 2015 Jan 21;10(1):e0115086. doi: 10.1371/journal.pone.0115086. PMID: 25607661; PMCID: PMC4301864.

One of largest critiques of this study was the data use for adolescents. It is common in research in adolescents to use normative values for many variables including GS, LBM, FMI, and any variable that is affected by growth and development. As the authors know, there is a lot of variability in the growth, development, and maturation of adolescents of the same age and sex. The outcomes of this study are misleading as absolute values of GS, FMI, and LBM are not comparable for adolescents. The data should be re-analyzed using GS percentile/z-score, FMI percentile/z-score, LBM percentile/z-score based on age, sex, height, etc. rather than absolute values. These normative values are available for each of these variables. There are likely Brazilian norms also.

                Grip strength:

                Bohannon, RW., Wang, YC., Bubela, D., Gershon, RC. Handgrip strength: A population-based study of norms and age trajectories for 3- to 17-year-olds. Pediatr Phys Ther. 2017; 29: 118-123.

                Wang YC, Bohannon RW, Li X, Sindhu B, Kapellusch J. Hand-Grip Strength: Normative Reference Values and Equations for Individuals 18 to 85 Years of Age Residing in the United States. J Orthop Sports Phys Ther. 2018 Sep;48(9):685-693.

Body comp (FMI, LBM):

                Weber, D. R., Moore, R. H., Leonard, M. B., & Zemel, B. S. (2013). Fat and lean BMI reference curves in children and adolescents and their utility in identifying excess adiposity compared with   BMI and percentage body fat. The American journal of clinical nutrition98(1), 49-56.

Wells, J. C., Williams, J. E., Chomtho, S., Darch, T., Grijalva-Eternod, C., Kennedy, K., ... & Fewtrell, M. S. (2012). Body-composition reference data for simple and reference techniques and a 4-component model: a new UK reference child. The American journal of clinical nutrition96(6), 1316-1326.

We appreciate the reviewer’s comment. We believe that linear growth has already been achieved and maturation has already been completed by these individuals, who were between 18 and 19 years old (the age range is only one year). And all analyzes performed were stratified by sex. Furthermore, we used the variables of body composition in the analyzes on a continuous basis, we did not classify the nutritional status of the individuals. From this, we do not think it is pertinent to change the analyses.

Figure 1 is very impressive

We appreciate the reviewer’s comment.

Results:

Line 160 – use ‘GS’ abbreviation

We appreciate the reviewer’s comment. We changed it as requested.

As noted in the abstract comments, it is unclear if FMI was included as a confounder in the LBM to GS analysis, or if LBM was included in the FMI to GS analysis. This issue presents again on lines 177-180. If those analyses include the listed confounders AND also adjust for the other primary variable (LBM or FMI), the authors should make this clear for the reader in the text.

We appreciate the reviewer’s comment. We changed it as requested.

Discussion:

The sentence on line 203-204 is repeated from the previous paragraph. The second sentence of paragraph 2 could start: ‘The association between LBM and GS was more intense…’

We appreciate the reviewer’s comment. We changed it as requested.

The sentence on lines 207-209 is confusing. Please consider re-writing.

We appreciate the reviewer’s comment. We have rewritten the sentence for better understanding.

Line 206 states that boys had less body fat distributed throughout their body. How can the authors determine segmented body fat distribution if they used a BodPod for determination of FM?

We appreciate the reviewer’s comment. As mentioned before, we do not study the body segment, but the entire body. And the IMF average of boys is half that of girls. Thus, we can say that boys have less fat than girls. However, we have modified the phrase for better understanding.

Line 210-211 – As noted previously, the complex physiology leading to increased LBM in males is more complex than just saying ‘because of their testosterone.’ I recommend making these kinds of statements more general.  

We appreciate the reviewer’s comment. We withdraw from the text, to make it more general, as requested.

The authors state on line 212 that ‘Evidence shows that males consume healthier and less ultra-processed foods than girls.’ The authors collected similar data. Does the author’s data support this?

We appreciate the reviewer’s comment. Yes, because the cited article was carried out with the same sample that was used in this article.

Line 215-218 should be deleted. The authors did not use normative values for GS, FMI, LBM. Thus, the outcomes of this study, and the interpretations made in these sentences are misleading and incomplete. Strength, LBM is relative to body size, maturational stage, and sex.

We appreciate the reviewer’s comment.  We deleted the paragraph as suggested.

Again, we welcome reviewers’ suggestions. We believe that they contributed to improving the article, which we would like to see published in the Nutrients.

Yours sincerely,

The authors.

Reviewer 2 Report

I have read with interest the paper entitled  "ARE FAT MASS AND LEAN MASS ASSOCIATED WITH 2 GRIP STRENGTH IN ADOLESCENTS?". The aim of this cross sectional study is to analyze the association between FMI and LBM with GS. This article meets the aims and the scope of the journal.

Nevertheless I have some major comments

Data collection:Please give further details for the procedure

Data analysis 1st paragraph:corrections are necessary, statistical terms are wrong :chi-square is a statistical test (inferential statistics). I suppose that the statistical test used is the Mann Whitney test and not the Wilkoxon test.

Table 2 :Further details are needed at least for adjusted analysis (statistic, SE)

Figures:Further details are needed for figures, especially for figure 1 

Author Response

Response letter

August 2, 2022

Article: ARE FAT MASS AND LEAN MASS ASSOCIATED WITH GRIP STRENGTH IN ADOLESCENTS?

Dear Editor,

We would like to thank you for reviewing and commenting on our article. Suggested changes are answered separately.

The following is a detailed description of how we addressed reviewers’ comments and suggestions.

Reviewer 2

Data collection: Please give further details for the procedure

We appreciate the reviewer’s comment. We changed it as requested.

Data analysis 1st paragraph: corrections are necessary, statistical terms are wrong: chi-square is a statistical test (inferential statistics). I suppose that the statistical test used is the Mann Whitney test and not the Wilkoxon test.

We appreciate the reviewer’s comment.  We changed it as requested.

Table 2: Further details are needed at least for adjusted analysis (statistic, SE)

We appreciate the reviewer’s comment. We changed it as requested.

Figures: Further details are needed for figures, especially for figure 1 

We appreciate the reviewer’s comment. We changed it as requested.

Again, we welcome reviewers’ suggestions. We believe that they contributed to improving the article, which we would like to see published in the Nutrients.

Yours sincerely,

The authors.

Round 2

Reviewer 1 Report

Thank you for addressing the comments. I see after reading your responses that I miss interpreted the ages of the participants in the study. I agree that age 18-19 likely have met full maturation. No other comments or suggestion. This is a very nice paper.